# V4D: 4D Convolutional Neural Networks for Video-level Representation Learning

**Shiwen Zhang, Sheng Guo, Weilin Huang**[*] **& Matthew R. Scott**
Malong Technologies, Shenzhen, China
Shenzhen Malong Artificial Intelligence Research Center, Shenzhen, China
`{shizhang,sheng,whuang,mscott}@malong.com`

**Limin Wang**
State Key Laboratory for Novel Software Technology, Nanjing University, China
`lmwang@nju.edu.cn`

## Abstract

Most existing 3D CNNs for video representation learning are clip-based methods, and thus do not consider video-level temporal evolution of spatio-temporal features. In this paper, we propose Video-level 4D Convolutional Neural Networks, referred as V4D, to model the evolution of long-range spatio-temporal representation with 4D convolutions, and at the same time, to preserve strong 3D spatio-temporal representation with residual connections. Specifically, we design a new 4D residual block able to capture inter-clip interactions, which could enhance the representation power of the original clip-level 3D CNNs. The 4D residual blocks can be easily integrated into the existing 3D CNNs to perform long-range modeling hierarchically. We further introduce the training and inference methods for the proposed V4D. Extensive experiments are conducted on three video recognition benchmarks, where V4D achieves excellent results, surpassing recent 3D CNNs by a large margin.

## 1 Introduction

3D convolutional neural networks (3D CNNs) and their variants (Ji et al., 2010; Tran et al., 2015; Carreira & Zisserman, 2017; Qiu et al., 2017; Wang et al., 2018b) provide a simple extension from 2D counterparts for video representation learning. However, due to practical issues such as memory consumption and computational cost, these models are mainly used for clip-level feature learning instead of learning from the whole video. The clip-based methods randomly sample a short clip (e.g., 32 frames) from a video for representation learning, and calculate prediction scores for each clip independently. The prediction scores of all clips are simply averaged to yield the video-level prediction. These clip-based models often ignore the video-level structure and long-range spatio-temporal dependency during training, as they only sample a small portion of the entire video. In fact, in some cases, it could be difficult to identify an action correctly by only using partial observation. Meanwhile, simply averaging the prediction scores of all clips could be sub-optimal during inference. To overcome this issue, Temporal Segment Network (TSN) (Wang et al., 2016) was proposed. TSN uniformly samples multiple clips from the entire video, and the average scores are used to guide back-propagation during training. Thus TSN is a video-level representation learning framework. However, the inter-clip interaction and video-level fusion in TSN is only performed at very late stage, which fails to capture finer temporal structures.

In this paper, we propose a general and flexible framework for video-level representation learning, called V4D. As shown in Figure 1, to model long-range dependency in a more efficient way, V4D is composed of two critical designs: (1) holistic sampling strategy, and (2) 4D convolutional interaction. We first introduce a video-level sampling strategy by uniformly sampling a sequence of short-term units covering the whole video. Then we model long-range spatio-temporal dependency by designing a unique 4D residual block. Specifically, we present a 4D convolutional operation to capture inter-clip

---

[*]The corresponding author.

interaction, which could enhance the representation power of the original clip-level 3D CNNs. The 4D residual blocks could be easily integrated into the existing 3D CNNs to perform long-range modeling hierarchically, which is more efficient than TSN. We also design a specific video-level inference algorithm for V4D. Finally, we verify the effectiveness of V4D on three video action recognition benchmarks, Mini-Kinetics (Xie et al., 2018), Kinetics-400 (Carreira & Zisserman, 2017) and Something-Something-V1 (Goyal et al., 2017). Our V4D achieves very competitive performance on the three benchmarks, and obtains evident performance improvement over its 3D counterparts.

## 2 RELATED WORKS

**Two-stream CNNs.** Two-stream architecture was originally proposed by (Simonyan & Zisserman, 2014), where one stream is used for learning from RGB images, and the other one is applied to model optical flow. The results produced by the two streams are then fused at later stages, yielding the final prediction. Two-stream CNNs have achieved impressive results on various video recognition tasks. However, the main limitation is that the computation of optical flow is highly expensive where parallel optimization is difficult to implment, with significant resource explored. Recent effort has been devoted to reducing the computational cost on modeling optical flow, such as (Dosovitskiy et al., 2015; Sun et al., 2018; Piergiovanni & Ryoo, 2018; Zhang et al., 2016). The two-stream design is a general framework to boost the performance of various CNN models, which is orthogonal to the proposed V4D.

**3D CNNs.** Recently, 3D CNNs have been proposed (Tran et al., 2015; Carreira & Zisserman, 2017; Wang et al., 2018a;b; Feichtenhofer et al., 2018). By considering a video as a stack of frames, it is natural to develop 3D convolutions applied directly on video sequence. However, 3D CNNs often introduce a large number of model parameters, which inevitably require a large amount of training data to achieve good performance. As reported in (Wang et al., 2018b; Feichtenhofer et al., 2018), recent experimental results on large-scale benchmark, likes Kinetics-400 (Carreira & Zisserman, 2017), show that 3D CNNs can surpass their 2D counterparts in many cases,and even can be on par with or better than the two-stream 2D CNNs. It is noteworthy that most of 3D CNNs are clip-based methods, which only explore a certain part of the holistic video.

**Long-term Modeling Framework.** Various long-term modeling frameworks have been developed for capturing more complex temporal structure for video-level representation learning. In (Laptev et al., 2008), video compositional models were proposed to jointly model local video events, where temporal pyramid matching was introduced with a bag-of-visual-words framework to compute long-term temporal structure. However, the rigid composition only works under defined constraints, e.g., prefixed duration and anchor points provided in time. A mainstream method is to process a continuous video sequence with recurrent neural networks Ng et al. (2015); Donahue et al. (2015), where 2D CNNs are used for frame-level feature extraction. Temporal Segment Network (TSN) (Wang et al., 2016) has been proposed to model video-level temporal information with a sparse sampling and aggregation strategy. TSN sparsely samples a set of frames from the whole video, and then the sampled frames are modelled by the same CNN backbone, which outputs a confident score for each frame. The output scores are averaged to generate final video-level prediction. TSN was originally designed for 2D CNNs, but it can be applied to 3D CNNs, which serves as one of the baselines in this paper. One of the main limitations of TSN is that it is difficult to model finer temporal structure due to the average aggregation. Temporal Relational Reasoning Network (TRN) (Zhou et al., 2018) was introduced to model temporal segment relation by encoding individual representation of each segment with relation networks. TRN is able to model video-level temporal order but lacks the capacity of capturing finer temporal structure. The proposed V4D can outperform these previous video-level learning methods on both appearance-dominated video recognition (e.g., on Kinetics) and motion-dominated video recognition (e.g., on Something-Something). It is able to model both short-term and long-term temporal structure with a unique design of 4D residual blocks.

## 3 VIDEO-LEVEL 4D COVOLUTIONAL NEURAL NETWORKS

In this section, we introduce new **Video-level 4D** Convolution Neural Networks, namely V4D, for video action recognition. This is the first attempt to design 4D convolutions for RGB-based video recognition. Previous methods, such as You & Jiang (2018); Choy et al. (2019), utilize 4D CNNs to

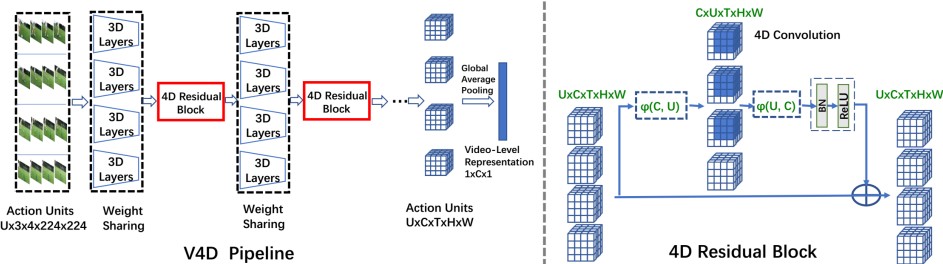

Figure 1: Video-level 4D Convolutional Neural Networks for video recognition.

process videos of point cloud by using 4D data as input. Instead, our V4D processes videos of RGB frames with input of 3D data. Existing 3D CNNs often take a short-term snippet as input, without considering the evolution of 3D spatio-temporal features for video-level representation. In Wang et al. (2018b); Yue et al. (2018); Liu et al. (2019), self-attention mechanism was developed to model non-local spatio-temporal features, but these methods were originally designed for clip-based 3D CNNs. It remains unclear how to incorporate such operations on holistic video representation, and whether such operations are useful for video-level representation learning. Our goal is to model 3D spatio-temporal features globally, which can be implemented in a higher dimension. In this work, we introduce new Residual 4D Blocks, which allow us to cast 3D CNNs into 4D CNNs for learning long-range interactions of the 3D features, resulting in a "time of time" video-level representation.

### 3.1 A VIDEO-LEVEL SAMPLING STRATEGY

To model meaningful video-level representation for action recognition, the input to the networks has to cover the holistic duration of a given video, and at the same time preserve short-term action details. A straightforward approach is to implement per-frame training of the networks yet this is not practical by considering the limit of computation resource. In this work, we uniformly divide the whole video into $U$ sections, and select a snippet from each section to represent a short-term action pattern, called "action unit". Then we have $U$ action units to represent the holistic action in a video. Formally, we denote the video-level input $V = \{A_1, A_2, ..., A_U\}$, where $A_i \in \mathbb{R}^{C \times T \times H \times W}$. During training, each action unit $A_i$ is randomly selected from each of the $U$ sections. During testing, the center of each $A_i$ locates exactly at the center of the corresponding section.

### 3.2 4D CONVOLUTIONS FOR LEARNING SPATIO-TEMPORAL INTERACTIONS

3D Convolutional kernels have been proposed, and are powerful to model short-term spatio-temporal features. However, the receptive fields of 3D kernels are often limited due to the small sizes of kernels, and pooling operations are applied to enlarge the receptive fields, resulting in a significant cost of information loss. This inspired us to develop new operations which are able to model both short- and long-term spatio-temporal representations simultaneously, with easy implementations and fast training. From this prospective, we propose 4D convolutions for better modeling the long-range spatio-temporal interactions.

Specifically, we denote the input to 4D convolutions as a tensor $V$ of size $(C, U, T, H, W)$, where $C$ is number of channel, $U$ is the number of action units (the 4-th dimension in this paper), $T, H, W$ are temporal length, height and width of an action unit. We omit the batch dimension for simplicity. By following the annotations provided in Ji et al. (2010), a pixel at position $(u, t, h, w)$ of the $j$th channel in the output is denoted as $o_j^{uthw}$, and a 4D convolution operation can be formulated as :

$$o_j^{uthw} = b_j + \sum_{c}^{C_{in}} \sum_{s=0}^{S-1} \sum_{p=0}^{P-1} \sum_{q=0}^{Q-1} \sum_{r=0}^{R-1} \mathcal{W}_{jc}^{spqr} v_c^{(u+s)(t+p)(h+q)(w+r)} \quad (1)$$

where $b_j$ is a bias term, $c$ is one of the $C_{in}$ input channels of the feature maps from input $V$, $S \times P \times Q \times R$ is the shape of 4D convolutional kernel, $\mathcal{W}_{jc}^{spqr}$ is the weight at the position $(s, p, q, r)$ of the kernel, corresponding to the $c$-th channel of the input feature maps and $j$-th channel of the output feature maps.

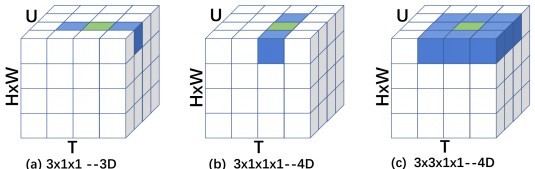

Figure 2: Implementation of 4D kernels, compared to3D kernel s. $U$ denotes the number of action units, with shape of $T, H, W$. Channel and batch dimensions are omitted for clarity. The kernels are colored in Blue, with the center of each kernel colored in Green.

Convolutional operation is linear, and the sequential sum operations in E.q. 1 are exchangeable. Thus we can generate E.q. 2, where the expression in the parentheses can be implemented by 3D convolutions, allowing us to implement 4D convolutions using 3D convolutions, while most deep learning libraries do not directly provide 4D convolutional operations.

$$o_j^{uthw} = b_j + \sum_{s=0}^{S-1}(\sum_{c}^{C_{in}}\sum_{p=0}^{P-1}\sum_{q=0}^{Q-1}\sum_{r=0}^{R-1}\mathcal{W}_{jc}^{spqr}v_c^{(u+s)(t+p)(h+q)(w+r)}) \tag{2}$$

With the 4D convolutional kernel, the short-term 3D features of an individual action unit and long-term temporal evolution of multiple action units can be modeled simultaneously in the 4D space. Compared to 3D convolutions, the proposed 4D convolutions are able to model videos in a more meaningful 4D feature space that enables it to learn more complicated interactions of long-range 3D spatio-temporal representation. However, 4D convolutions inevitably introduce more parameters and computation cost. For example, a 4D convolutional kernel of $k \times k \times k \times k$ employs $k$ times more parameters than a 3D kernel of $k \times k \times k$. In practice, we explore $k \times k \times 1 \times 1$ and $k \times 1 \times 1 \times 1$ kernels, to reduce the parameters and avoid the risk of overfitting. The implementation of different kernels is shown in Figure 2.

### 3.3 VIDEO-LEVEL 4D CNN ARCHITECTURE

In this section, we demonstrate that our 4D convolutions can be integrated into existing CNN architecture for action recognition. To fully utilize current state-of-the-art 3D CNNs, we propose a new Residual 4D Convolution Block, by designing a 4D convolution in the residual structure introduced in (He et al., 2016). This allows it to aggregate both short-term 3D features and long-term evolution of the spatio-temporal representations for video-level action recognition. Specifically, we define a permutation function $\varphi_{(d_i,d_j)} : M^{d_1 \times ... \times d_i \times ... \times d_j \times ... \times d_n} \mapsto M^{d_1 \times ... \times d_j \times ... \times d_i \times ... \times d_n}$, which permutes dimension $d_i$ and $d_j$ of a tensor $M \in \mathbb{R}^{d_1 \times ... \times d_n}$. The Residual 4D Convolution Block can be formulated as:

$$\mathcal{Y}_{3D} = \mathcal{X}_{3D} + \varphi_{(U,C)}(\mathcal{F}_{4D}(\varphi_{(C,U)}(\mathcal{X}_{3D}); \mathcal{W}_{4D})) \tag{3}$$

where $\mathcal{F}_{4D}(\mathcal{X}; \mathcal{W}_{4D})$ is the introduced 4D convolution. $\mathcal{X}_{3D}, \mathcal{Y}_{3D} \in \mathbb{R}^{U \times C \times T \times H \times W}$, and $U$ is merged into batch dimension so that $\mathcal{X}_{3D}, \mathcal{Y}_{3D}$ can be directly processed by standard 3D CNNs. Note that we employ $\varphi$ to permute the dimensions of $\mathcal{X}_{3D}$ from $U \times C \times T \times H \times W$ to $C \times U \times T \times H \times W$ so that it can be processed by 4D convolutions. Then the output of 4D convolution is permuted reversely to 3D form so that the output dimensions are consistent with $\mathcal{X}_{3D}$. Batch Normalization (Ioffe & Szegedy, 2015) and ReLU activation (Nair & Hinton, 2010) are then applied. The detailed structure is described in Figure 1.

Theoretically, any 3D CNN structure can be cast to 4D CNNs by integrating our 4D Convolutional Blocks. As shown in previous works (Zolfaghari et al., 2018; Xie et al., 2018; Wang et al., 2018b; Feichtenhofer et al., 2018), higher performance can be obtained by applying 2D convolutions at lower layers and 3D convolutions at higher layers of the 3D networks. In our framework, we utilize the "Slow-path" introduced in Feichtenhofer et al. (2018) as our backbone, denoted as I3D-S. Although the original "Slowpath" is designed for ResNet50, we can extend it to I3D-S ResNet18 for further experiments. The detailed structure of our 3D backbone is shown in Table 1.

| layer | I3D-S ResNet18 | I3D-S ResNet50 | output size |
|---|---|---|---|
| $conv_1$ | $1\times7\times7$, 64, stride 1, 2, 2 | $1\times7\times7$, 64, stride 1, 2, 2 | $4\times112\times112$ |
| $res_2$ | $\begin{bmatrix} 1\times3\times3,\ 64 \\ 1\times3\times3,\ 64 \end{bmatrix}\times2$ | $\begin{bmatrix} 1\times1\times1,\ 64 \\ 1\times3\times3,\ 64 \\ 1\times1\times1,\ 256 \end{bmatrix}\times3$ | $4\times56\times56$ |
| $res_3$ | $\begin{bmatrix} 1\times3\times3,\ 128 \\ 1\times3\times3,\ 128 \end{bmatrix}\times2$ | $\begin{bmatrix} 1\times1\times1,\ 128 \\ 1\times3\times3,\ 128 \\ 1\times1\times1,\ 512 \end{bmatrix}\times4$ | $4\times28\times28$ |
| $res_4$ | $\begin{bmatrix} 3\times3\times3,\ 256 \\ 3\times3\times3,\ 256 \end{bmatrix}\times2$ | $\begin{bmatrix} 3\times1\times1,\ 256 \\ 1\times3\times3,\ 256 \\ 1\times1\times1,\ 1024 \end{bmatrix}\times6$ | $4\times14\times14$ |
| $res_5$ | $\begin{bmatrix} 3\times3\times3,\ 512 \\ 3\times3\times3,\ 512 \end{bmatrix}\times2$ | $\begin{bmatrix} 3\times1\times1,\ 512 \\ 1\times3\times3,\ 512 \\ 1\times1\times1,\ 2048 \end{bmatrix}\times3$ | $4\times7\times7$ |
| | global average pool, fc | | $1\times1\times1$ |

Table 1: We use I3D-Slowpath from (Feichtenhofer et al., 2018) as our backbone. The output size of an example is shown in the right column, where the input has a size of $4\times224\times224$. No temporal degenerating is performed in this structure.

## 3.4 Training and Inference

**Training.** As shown in Figure 1, the convolutional part of the networks is composed of 3D convolution layers and the proposed Residual 4D Blocks. Each action unit is trained individually and in parallel in the 3D convolution layers, which share the same parameters. The 3D features computed from the action units are then fed to the Residual 4D Block, where the long-term temporal evolution of the consecutive action units can be modeled. Finally, global average pooling is computed on the sequence of all action units to form the final video-level representation.

**Inference.** Given $U$ action units $\{A_1, A_2, ..., A_U\}$ of a video, we denote $U_{train}$ as the number of action units for training and $U_{infer}$ as the number of action units for inference. $U_{train}$ and $U_{infer}$ are usually different because computation resource is limited in training, but high accuracy is encouraged in inference. We develop a new video-level inference method, which is described in Algorithm 1. The 3D convolutional layers are denote as $N_{3D}$, followed by the proposed 4D Blocks, $N_{4D}$.

---

**Algorithm 1:** V4D Inference.

| **Networks** | : The structure of networks is divided into two sub-networks by the first 4D Block, namely $N_{3D}$ and $N_{4D}$. |
|---|---|
| **Input** | : $U_{infer}$ action units from a holistic video: $\{A_1, A_2, ..., A_{U_{infer}}\}$. |
| **Output** | : The video-level prediction. |

**V4D Inference :**

1 $\{A_1, A_2, ..., A_{U_{infer}}\}$ are fed into $N_{3D}$, generating intermediate feature maps for each unit $\{F_1, F_2, ..., F_{U_{infer}}\}, F_i \in \mathbb{R}^{C\times T\times H\times W}$;

2 For the $U_{infer}$ intermediate features, we equally divide them into $U_{train}$ sections. Then we select one unit from each section $F_{sec_i}$ and combine these $U_{train}$ units into a video-level intermediate representation $F^{video} = (F_{sec_1}, F_{sec_2}, ..., F_{sec_{U_{train}}})$. These video-level representations form a new set $\{F_1^{video}, F_2^{video}, ..., F_{U_{combined}}^{video}\}$, where $U_{combined} = (U_{infer}/U_{train})^{U_{train}}, F_i^{video} \in \mathbb{R}^{U_{train}\times C\times T\times H\times W}$;

3 Each $F_i^{video}$ in set $\{F_1^{video}, F_2^{video}, ..., F_{U_{combined}}^{video}\}$ are processed by $N_{4D}$ to form a set of prediction scores, $\{P_1, P_2, ..., P_{U_{combined}}\}$;

4 $\{P_1, P_2, ..., P_{U_{combined}}\}$ are averaged to give the final video-level prediction.

---

## 3.5 Discussion

We further demonstrate that the proposed V4D can be considered as a 4D generalization of a number of recent widely-applied methods, which may partially explain why our V4D works practically well on learning meaningful video-level representation.

**Temporal Segment Network**. Our V4D is closely related to Temporal Segment Network (TSN). TSN was originally designed for 2D CNN, but it can be directly applied to 3D CNN to model

video-level representation. TSN also employs a video-level sampling strategy with each action unit named "segment". During training, each segment is calculated individually and the prediction scores after the fully-connected layer are then averaged. Since the fully-connected layer is a linear classifier, it is mathematically identical to calculating the average before the fully-connected layer (similar to our global average pooling) or after the fully-connected layer (similar to TSN). Thus our V4D can be considered as 3D CNN + TSN when all parameters in the 4D Blocks are set to 0.

**Dilated Temporal Convolution**. One special form of 4D convolution kernel, $k \times 1 \times 1 \times 1$, is closely related to Temporal Dilated Convolution (Lea et al., 2016). The input tensor $V$ can be considered as a $(C, U \times T, H, W)$ tensor when all action units are concatenated along the temporal dimension. In this case, the $k \times 1 \times 1 \times 1$ 4D convolution can be considered as a dilated 3D convolution kernel of $k \times 1 \times 1$ with a dilation of $T$ frames. Note that the $k \times 1 \times 1 \times 1$ kernel is just the simplest form of our 4D convolutions, while our V4D architecture can utilize more complex kernels and thus can be more meaningful for learning stronger video representation. Furthermore, our 4D Blocks utilize residual connections, ensuring that both long-term and short-term representation can be learned jointly. Simply applying the dilated convolution might discard the short-term fine-grained features.

# 4 EXPERIMENTS

## 4.1 DATASETS

We conduct experiments on three standard benchmarks: Mini-Kinetics (Xie et al., 2018), Kinetics-400 (Carreira & Zisserman, 2017), and Something-Something-v1 (Goyal et al., 2017). Mini-kinetics dataset covers 200 action classes, and is a subset of Kinetics-400. Since some videos are no longer available for Kinetics dataset, our version of Kinetics-400 contains 240,436 and 19,796 videos in the training subset and validation subset, respectively. Our version of Mini-kinetics contains 78,422 videos for training, and 4,994 videos for validation. Each video has around 300 frames. Something-Something-v1 contains 108,499 videos totally, with 86,017 for training, 11,522 for validation, and 10,960 for testing. Each video has 36 to 72 frames.

## 4.2 ABLATION STUDY ON MINI-KINETICS

We use pre-trained weights from ImageNet to initialize the model. For training, we adapt the holistic sampling strategy mentioned in section 3.1. We uniformly divide the whole video into $U$ sections, and randomly select a clip of 32 frames from each section. For each clip, by following the sampling strategy in Feichtenhofer et al. (2018), we uniformly sample 4 frames with a fixed stride of 8 to form an action unit. We will study the impact of $U$ in the following experiments. We first resize each frame to $320 \times 256$, and then randomly cropping is applied as Wang et al. (2018b). Then the cropped region is further resized to $224 \times 224$. We utilize a SGD optimizer with an initial learning rate of 0.01, weight decay is set to $10^{-5}$ with a momentum of 0.9. The learning rate drops by 10 at epoch 35, 60, 80, and the model is trained for 100 epochs in total.

To make a fair comparison, we use spatial fully convolutional testing by following Wang et al. (2018b); Yue et al. (2018); Feichtenhofer et al. (2018). We sample 10 action units evenly from a full-length video, and crop $256 \times 256$ regions to spatially cover the whole frame for each action unit. Then we apply the proposed V4D inference. Note that, for the original TSN, 25 clips and 10-crop testing are used during inference. To make a fair comparison between I3D and our V4D, we instead apply this 10 clips and 3-crop inference strategy for TSN.

**Results.** To verify the effectiveness of V4D, we compare it with the clip-based method I3D-S, and video-based method TSN+3D CNN. To compensate the extra parameters introduced by 4D blocks, we add a $3 \times 3 \times 3$ residual block at res4 for I3D-S for a fair comparison, denoted as I3D-S ResNet18++. As shown in Table 2a, by using 4 times less frames than I3D-S during inference and with less parameters than I3D-S ResNet18++, V4D still obtain a 2.0% higher top-1 accuracy than I3D-S. Comparing with the state-of-the-art video-level method TSN+3D CNN, V4D significantly outperforms it by 2.6% top-1 accuracy, with the same protocols used in training and inference.

**4D Convolution Kernels.** As mentioned, our 4D convolution kernels can use 3 typical forms: $k \times 1 \times 1 \times 1$, $k \times k \times 1 \times 1$ and $k \times k \times k \times k$. In this experiment, we set $k = 3$ for simplicity, and apply a single 4D block at the end of res4 in I3D-S ResNet18. As shown in Table 2c, V4D with

| model | $T_{train} \times U_{train}$ | $T_{infer} \times U_{infer} \times$ #crop | top-1 | top5 | parameters |
|---|---|---|---|---|---|
| I3D-S ResNet18 | $4 \times 1$ | $4 \times 10 \times 3$ | 72.2 | 91.2 | 32.3M |
| I3D-S ResNet18 | $16 \times 1$ | $16 \times 10 \times 3$ | 73.4 | 91.1 | 32.3M |
| I3D-S ResNet18++ | $16 \times 1$ | $16 \times 10 \times 3$ | 73.6 | 91.5 | 34.1M |
| TSN+I3D-S ResNet18 | $4 \times 4$ | $4 \times 10 \times 3$ | 73.0 | 91.3 | 32.3M |
| V4D ResNet18 | $4 \times 4$ | $4 \times 10 \times 3$ | 75.6 | 92.7 | 33.1M |

(a) Effectiveness of V4D. $T$ represents temporal length of each action unit. $U$ represents the number of action units.

| model | input size | flops |
|---|---|---|
| I3D-S ResNet18 | $16 \times 256 \times 256$ | 55.1G |
| TSN+I3D-S ResNet18 | $4 \times 4 \times 256 \times 256$ | 55.1G |
| V4D ResNet18 | $4 \times 4 \times 256 \times 256$ | 58.8G |

(b) Forward flops of previous works and V4D. One 4D block at res3 and one at res4 for V4D.

| model | form of 4D kernel | top-1 | top5 |
|---|---|---|---|
| I3D-S ResNet18 | - | 72.2 | 91.2 |
| TSN+I3D-S ResNet18 | - | 73.0 | 91.3 |
| V4D ResNet18 | $3 \times 1 \times 1 \times 1$ | 73.8 | 92.0 |
| V4D ResNet18 | $3 \times 3 \times 1 \times 1$ | 74.5 | 92.4 |
| V4D ResNet18 | $3 \times 3 \times 3 \times 3$ | 74.7 | 92.5 |

(c) Different Forms of 4D Convolution Kernel.

| model | 4D kernel | top-1 | top5 |
|---|---|---|---|
| I3D-S ResNet18 | - | 72.2 | 91.2 |
| TSN+I3D-S ResNet18 | - | 73.0 | 91.3 |
| V4D ResNet18 | 1 at res3 | 74.2 | 92.3 |
| V4D ResNet18 | 1 at res4 | 74.5 | 92.4 |
| V4D ResNet18 | 1 at res5 | 73.6 | 91.4 |
| V4D ResNet18 | 1 at res3, 1 at res4 | 75.6 | 92.7 |

(d) Position and Number of 4D Blocks.

| model | $U_{train}$ | top-1 | top5 |
|---|---|---|---|
| I3D-S ResNet18 | 1 | 72.2 | 91.2 |
| TSN+I3D-S ResNet18 | 4 | 73.0 | 91.3 |
| V4D ResNet18 | 3 | 74.3 | 92.2 |
| V4D ResNet18 | 4 | 74.5 | 92.4 |
| V4D ResNet18 | 5 | 74.5 | 92.3 |
| V4D ResNet18 | 6 | 74.6 | 92.5 |

(e) Effect of $U_{train}$.

Table 2: **Ablations** on Mini-Kinetics, with top-1 and top-5 classification accuracy (%).

$3 \times 3 \times 3 \times 3$ kernel can achieve the highest performance. However, by considering the trade-off between model parameters and performance, we use the kernel of $3 \times 3 \times 1 \times 1$ in the following experiments.

**On 4D Blocks.** We evaluate the impact of position and number of 4D Blocks for our V4D. We investigate the performance of V4D by using one $3 \times 3 \times 1 \times 1$ 4D block at res3, res4 or res5. As shown in Table 2d, a higher accuracy can be obtained by applying the 4D block at res3 or res4, indicating that the merged long-short term features of the 4D block need to be further refined by 3D convolutions to generate more meaningful representation. Furthermore, inserting one 4D block at res3 and one at res4 can achieve a higher accuracy.

**Number of Action Units $U$.** We further evaluate our V4D by using different numbers of action units for training, with different values of hyperparameter $U$. In this experiment, one $3 \times 3 \times 1 \times 1$ Residual 4D block is added at the end of res4 of ResNet18. As shown in Table 2e, $U$ does not have a significant impact to the performance, which suggests that: (1) V4D is a video-level feature learning model, and is robust against the number of short-term units; (2) an action generally does not contain many stages, and thus increasing $U$ is not helpful. In addition, increasing the number of action units means that the 4-th dimension is increased, requiring a larger 4D kernel to cover the long-range evolution of spatio-temporal representation.

**With state-of-the-art.** We compare the results on Mini-Kinetics. 4D Residual Blocks are added into every other 3D residual blocks in res3 and res4. With much fewer frames utilized during training and inference, our V4D ResNet50 achieves a higher accuracy than all reported results, which is even higher than 3D ResNet101 having 5 compact Generalized Non-local Blocks. Note that our V4D ResNet18 can achieve a higher accuracy than 3D ResNet50, which further verify the effectiveness of our V4D structure.

| Model | Backbone | $T_{train} \times U_{train}$ | $T_{infer} \times U_{infer} \times$ #crop | top-1 | top5 |
|---|---|---|---|---|---|
| S3D (Xie et al., 2018) | S3D Inception | $64 \times 1$ | N/A | 78.9 | - |
| I3D (Yue et al., 2018) | 3D ResNet50 | $32 \times 1$ | $32 \times 10 \times 3$ | 75.5 | 92.2 |
| I3D (Yue et al., 2018) | 3D ResNet101 | $32 \times 1$ | $32 \times 10 \times 3$ | 77.4 | 93.2 |
| I3D+NL (Yue et al., 2018) | 3D ResNet50 | $32 \times 1$ | $32 \times 10 \times 3$ | 77.5 | 94.0 |
| I3D+CGNL (Yue et al., 2018) | 3D ResNet50 | $32 \times 1$ | $32 \times 10 \times 3$ | 78.8 | 94.4 |
| I3D+NL (Yue et al., 2018) | 3D ResNet101 | $32 \times 1$ | $32 \times 10 \times 3$ | 79.2 | 93.2 |
| I3D+CGNL (Yue et al., 2018) | 3D ResNet101 | $32 \times 1$ | $32 \times 10 \times 3$ | 79.9 | 93.4 |
| V4D(Ours) | V4D ResNet18 | $4 \times 4$ | $4 \times 10 \times 3$ | 75.6 | 92.7 |
| V4D(Ours) | V4D ResNet50 | $4 \times 4$ | $4 \times 10 \times 3$ | 80.7 | 95.3 |

Table 3: Results on Mini-Kinetics. $T$ - temporal length of action unit. $U$ - number of action units.

## 4.3 RESULTS ON KINETICS

We further conduct experiments on large-scale video recognition benchmark, Kinetics-400, to evaluate the capability of our V4D. To make a fair comparison, we utilize ResNet50 as backbone for V4D. The training and inference sampling strategy is identical to previous section, except that each action unit now contains 8 frames instead of 4. We set $U = 4$ so that there are $8 \times 4$ frames in total for training. Due to the limit of computation resource, we train the model in multiple stages. We first train the 3D ResNet50 backbone with 8-frame inputs. Then we load the 3D ResNet50 weights to V4D ResNet50, with all 4D Blocks fixed to zero. The V4D ResNet50 is then fine-tuned with $8 \times 4$ input frames. Finally, we optimize all 4D Blocks, and train the V4D with $8 \times 4$ frames. As shown in Table 4, our V4D achieves competitive results on Kinetics-400 benchmark.

| Model | Backbone | top-1 | top-5 |
|---|---|---|---|
| ARTNet with TSN (Wang et al., 2018a) | ARTNet ResNet18 | 70.7 | 89.3 |
| ECO (Zolfaghari et al., 2018) | BN-Inception+3D ResNet18 | 70.0 | 89.4 |
| S3D-G (Xie et al., 2018) | S3D Inception | 74.7 | 93.4 |
| Nonlocal Network (Wang et al., 2018a) | 3D ResNet50 | 76.5 | 92.6 |
| SlowFast (Feichtenhofer et al., 2018) | SlowFast ResNet50 | 77.0 | 92.6 |
| I3D(Carreira & Zisserman, 2017) | I3D Inception | 72.1 | 90.3 |
| Two-stream I3D(Carreira & Zisserman, 2017) | I3D Inception | 75.7 | 92.0 |
| I3D-S(Feichtenhofer et al., 2018) | Slow pathway ResNet50 | 74.9 | 91.5 |
| V4D(Ours) | V4D ResNet50 | 77.4 | 93.1 |

Table 4: Comparison with state-of-the-art on Kinetics.

## 4.4 RESULTS ON SOMETHING-SOMETHING-V1

Something-Something dataset focuses on modeling temporal information and motion. The background is much cleaner than Kinetics but the motions of action categories are more complicated. Each video contains one single and continuous action with clear start and end on temporal dimension.

| Model | Backbone | top-1 |
|---|---|---|
| MultiScale TRN (Zhou et al., 2018) | BN-Inception | 34.4 |
| ECO (Zolfaghari et al., 2018) | BN-Inception+3D ResNet18 | 46.4 |
| S3D-G (Xie et al., 2018) | S3D Inception | 45.8 |
| Nonlocal Network+GCN (Wang & Gupta, 2018) | 3D ResNet50 | 46.1 |
| TrajectoryNet (Zhao et al., 2018) | S3D ResNet18 | 47.8 |
| V4D(Ours) | V4D ResNet50 | 50.4 |

Table 5: Comparison with state-of-the-art on Something-Something-v1.

**Results.** As shown in Table 4.4, our V4D achieves competitive results on the Something-Something-v1. We use V4D ResNet50 pre-trained on Kinetics for experiments. **Temporal Order.** As shown in Xie et al. (2018), the performance can drop considerably by reversing the temporal order of short-term 3D features, suggesting that 3D CNNs can learn strong temporal order information. We further conduct experiments by reversing the frames within each action unit or reversing the sequence of action units, where the top-1 accuracy drops considerably by 50.4%→17.2% and 50.4%→20.1% respectively, indicating that our V4D can capture both long-term and short-term temporal order.

## 5 CONCLUSIONS

We have introduced new Video-level 4D Convolutional Neural Networks, namely V4D, to learn strong temporal evolution of long-range spatio-temporal representation, as well as retaining 3D features with residual connections. In addition, we further introduce the training and inference methods for our V4D. Experiments were conducted on three video recognition benchmarks, where our V4D achieved the state-of-the-art results.

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

# A    APPENDIX

## A.1    EXTENDED EXPERIMENTS ON LARGE-SCALE UNTRIMMED VIDEO RECOGNITION

In order to check the generalization ability of our proposed V4D, we also conduct experiments for untrimmed video classification. To be specific, we choose ActivityNet v1.3 Heilbron et al. (2015), which is a large-scale untrimmed video dataset, containing videos of 5 to 10 minutes and typically large time lapses of the videos are not related with any activity of interest. We adopt V4D ResNet50 to compare with previous works. During inference, Multi-scale Temporal Window Integration is applied following (Wang et al., 2016). The evaluation metric is mean average precision (mAP) for action recognition. Note that only RGB modality is used as input.

| Model | Backbone | mAP |
|---|---|---|
| TSN Wang et al. (2016) | BN-Inception | 79.7 |
| TSN Wang et al. (2016) | Inception V3 | 83.3 |
| TSN-Top3 Wang et al. (2016) | Inception V3 | 84.5 |
| V4D(Ours) | V4D ResNet50 | 88.9 |

Table 6: Comparison with state-of-the-art on ActivityNet v1.3.

## A.2    VISUALIZATION

We implement 3D CAM based on Zhou et al. (2016), which was originally implemented for 2D cases. Generally, class activation maps (CAM) imply which areas are most important for classification. Here we show some random visualization results from validation set of Mini-Kinetics, where TSN + I3D-S ResNet18 generates wrong prediction while V4D ResNet18 generates correct prediction. The original RGB frames are shown in the first row. The second row shows the CAMs of TSN + I3D-S ResNet18. The third row shows the CAMs of V4D ResNet18.

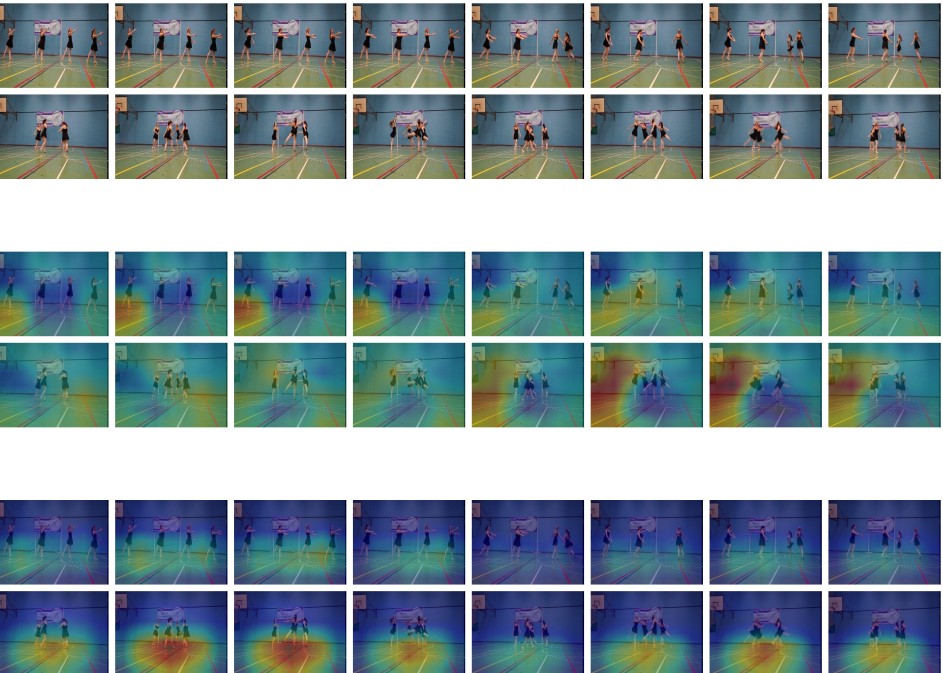

Figure 3:

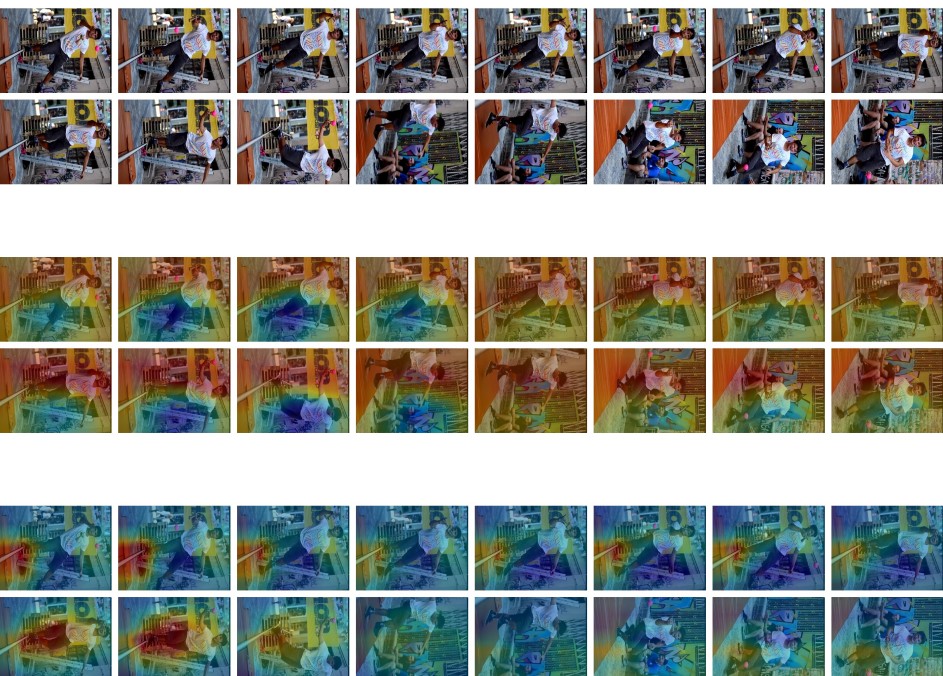

Figure 4:

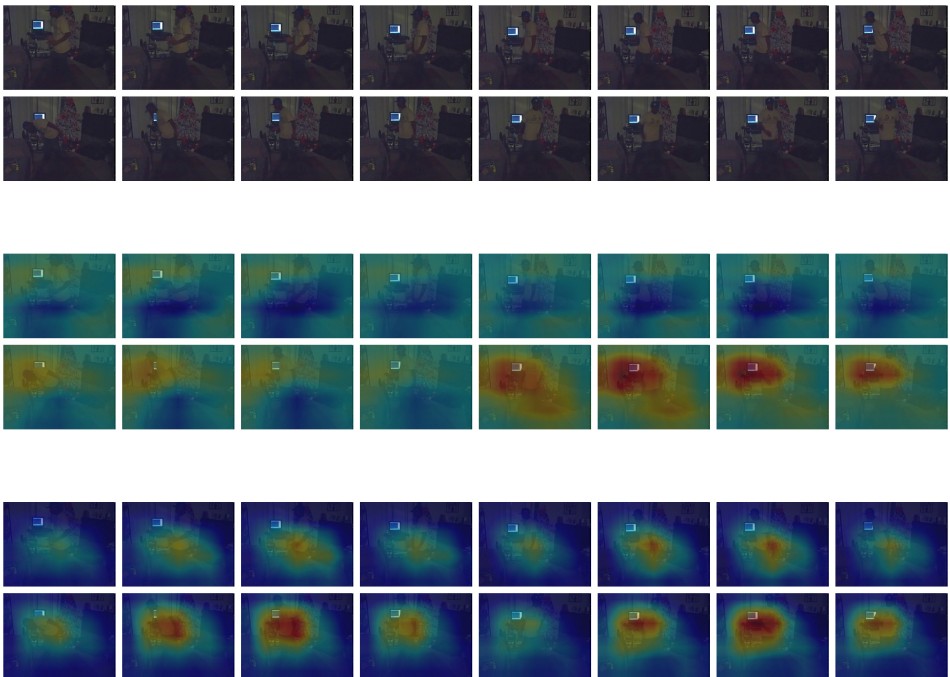

Figure 5:

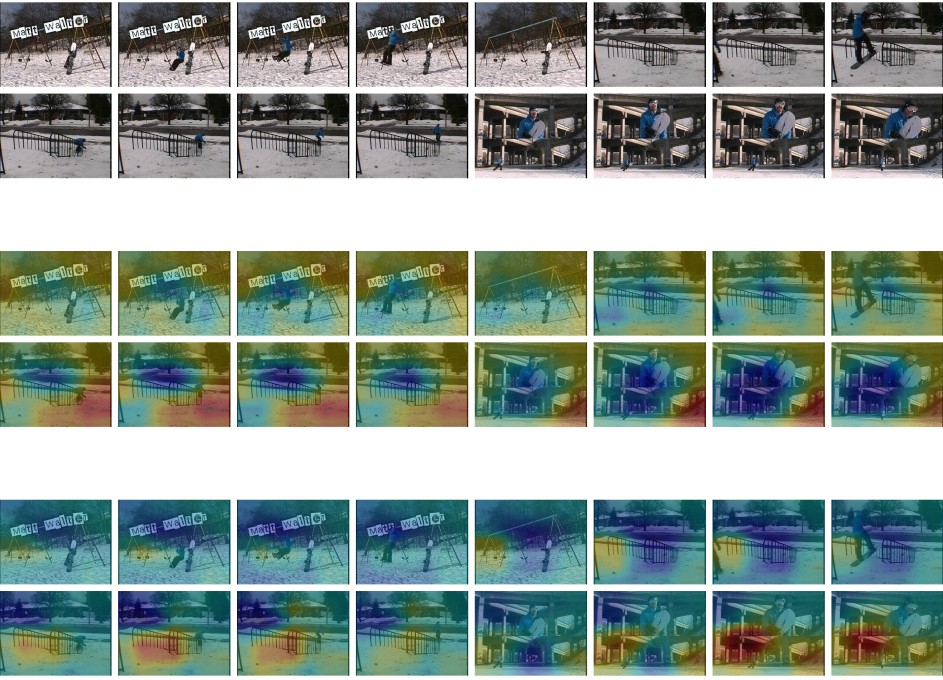

Figure 6:

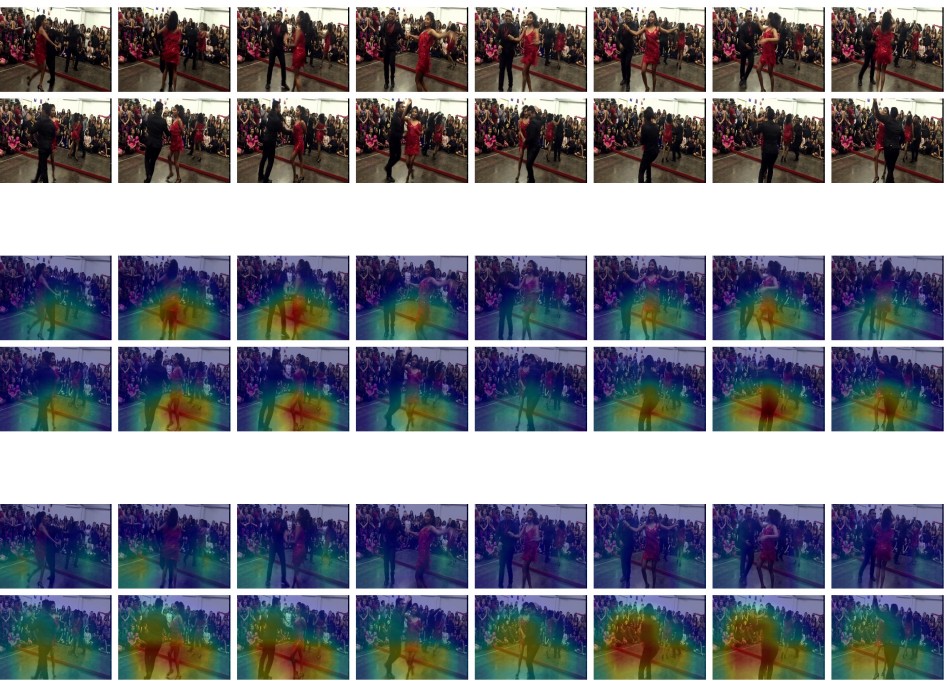

Figure 7:

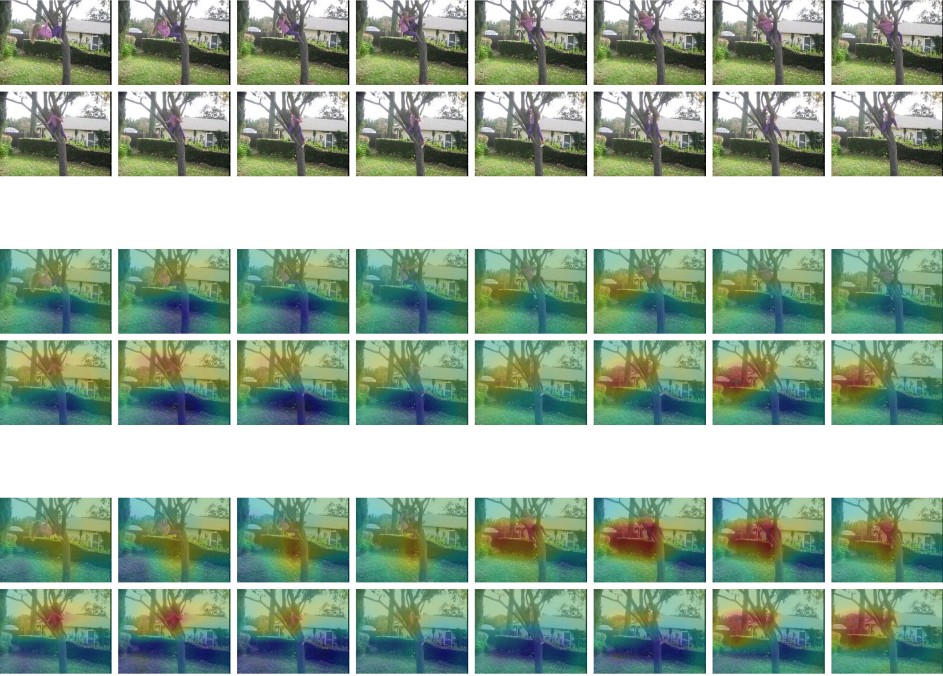

Figure 8:

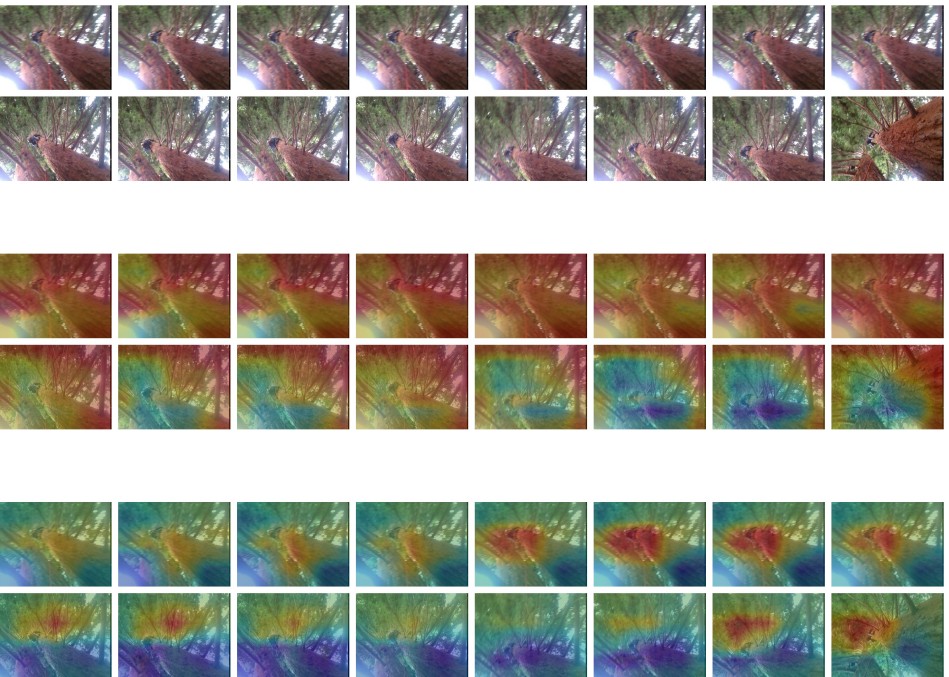

Figure 9:

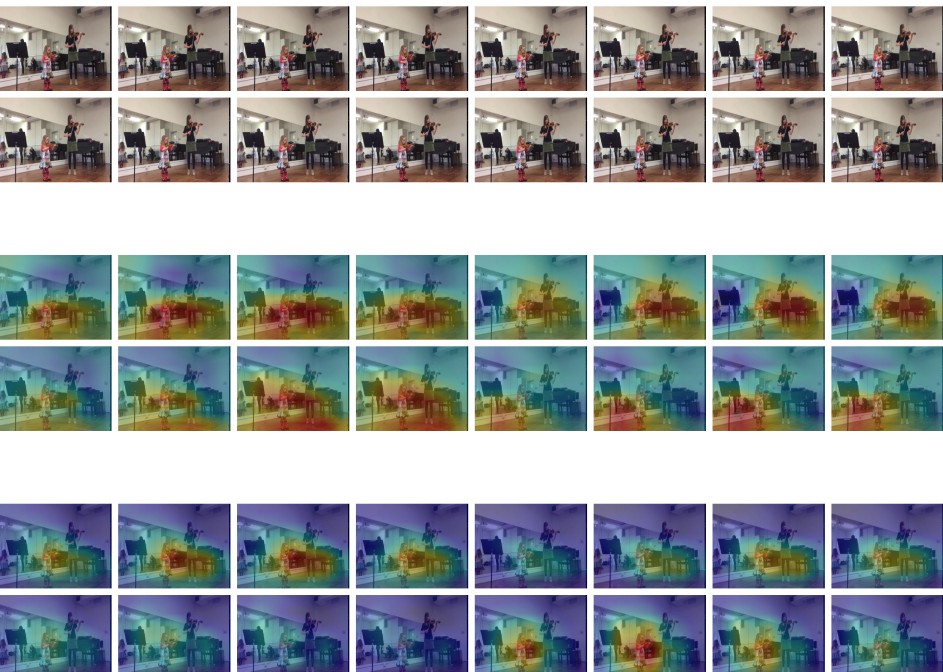

Figure 10:

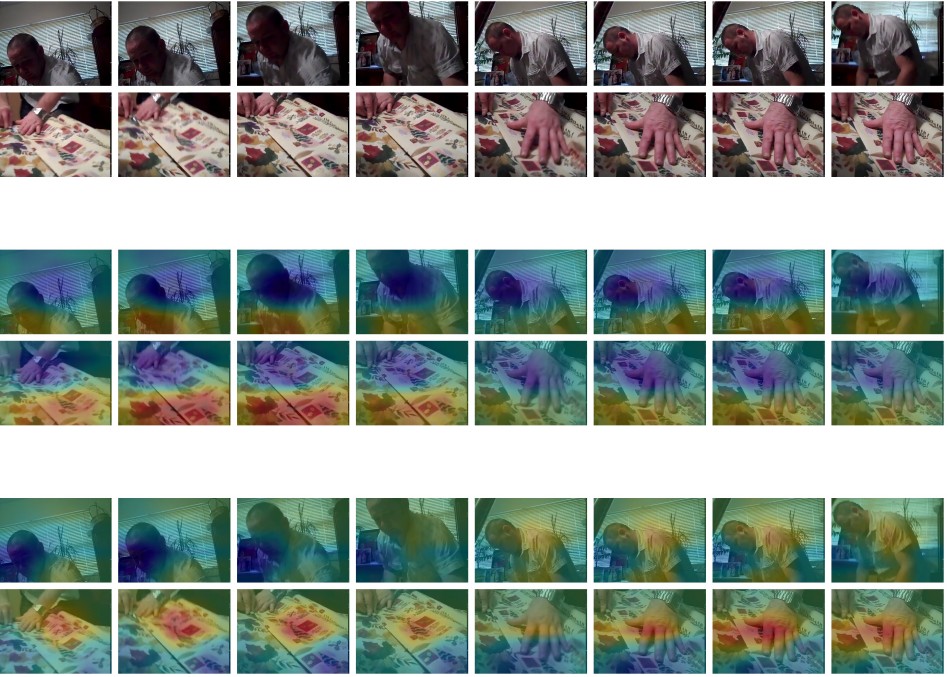

Figure 11:

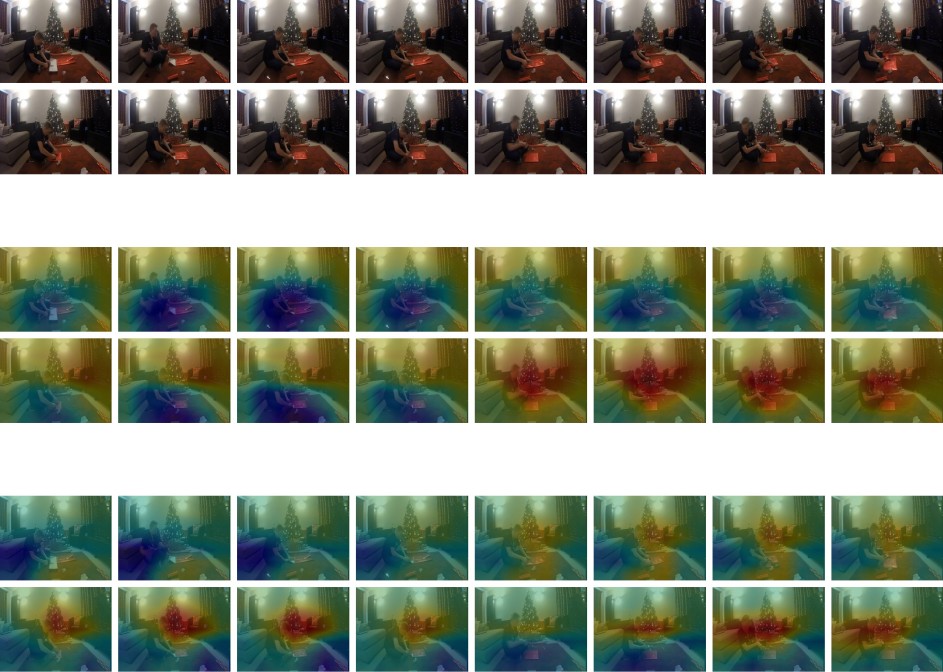

Figure 12:

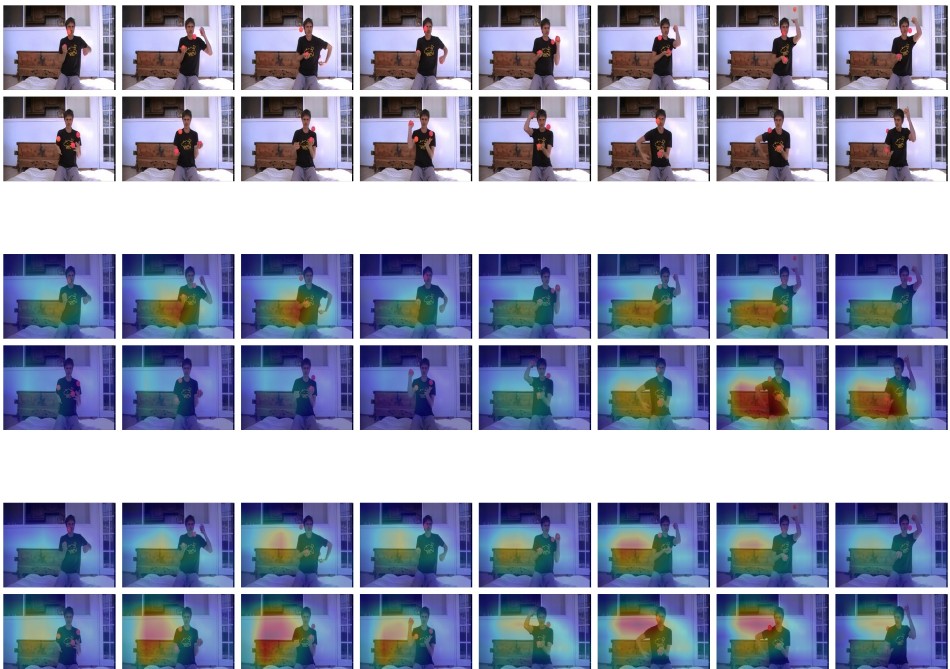

Figure 13:

