# OpenReview forum: "V4D: 4D Convolutional Neural Networks for Video-level Representation Learning"
_ICLR.cc/2020/Conference — Accept (Poster)_

### Official Review · AnonReviewer2 · 2019-10-23
**Official Blind Review #2**

**Rating:** 6

**Review:**

[Summary]
The paper presents a video classification framework that employs 4D convolution to capture longer term temporal structure than the popular 3D convolution schemes. This is achieved by treating the compositional space of local 3D video snippets as an individual dimension where an individual convolution is applied. The 4D convolution is integrated in resnet blocks and implemented via first applying 3D convolution to regular spatio-temporal video volumes and then the compositional space convolution, to leverage existing 3D operators. Empirical evaluation on three benchmarks against other baselines suggested the advantage of the proposed method.

[Decision]
Overall, the paper addresses an important problem in computer vision (video action recognition) with an interesting. I found the motivation and solution are reasonable (despite some questions pending more elaboration), and results also look promising, thus give it a weak accept (conditional on the answers though).

[Comments]
At the conceptual level, the idea of jointly modeling local video events is not novel, and can date back to at least ten years ago in the paper “Learning realistic human actions from movies”, where the temporal pyramid matching was combined with the bag-of-visual-words framework to capture long-term temporal structure. The problem with this strategy is that the rigid composition only works for actions that can be split into consecutive temporal parts with prefixed duration and anchor points in time, which is clearly challenged by many works later when more complicated video events are studied. It seems to me that the proposed framework also falls in this category, with a treatment from deep learning. It is definitely worth some discussion on this path.

That said, I would like to see more analysis on the behavior of the proposed method under various interesting cases not tested yet. Despite the claim that the proposed method can capture long-term video patterns, the static compositional nature seems to work best for activities with well-defined local events and clear temporal boundaries. These assumptions hold mostly true for the three datasets used in the experiment, and also are suggested by results in table 2(e), where 3 parts are necessary to achieve optimal results. How does the proposed method perform in more complicated tasks such as
- action detection or localization (e.g., in benchmarks JHMDB or UCF101-24).
- complex video event modeling (e.g., recognizing activities in extended video of TRECVID).
Will it still be more favorable than other concerning baselines?

Besides, on the computation side, it would be complexity, an explicit comparison of complexity makes it easier to evaluate the performance when compared to other state-of-the-art methods.

[Area to improve]
Better literature review to reflect the relevant previous video action recognitions, especially those on video compositional models.
Proof reading - The word in the title should be “Convolutional”, right?

**Experience Assessment:**

I have published one or two papers in this area.

**Review Assessment: Checking Correctness Of Derivations And Theory:**

I assessed the sensibility of the derivations and theory.

**Review Assessment: Checking Correctness Of Experiments:**

I assessed the sensibility of the experiments.

**Review Assessment: Thoroughness In Paper Reading:**

I read the paper at least twice and used my best judgement in assessing the paper.

---

> ### Author Response · Authors · 2019-11-13
> **Response for Reviewer#2**
>
> Thank you for your comments and suggestions. We will address the issues you mentioned.
>
>
> 1.	Thank you for the insightful suggestion. We now have added related work about video compositional methods in section 2.3 in the second version of the paper.
>
>
> 2.   In the original version of the paper, all experiments are conducted on trimmed video classification datasets. Although most papers in this field only report results on the trimmed video datasets, we do agree that more complicate cases should be tested. Additionally, we evaluated our V4D for untrimmed video classification on ActivityNet v1.3, which contains videos of 5 to 10 minutes and typically large time lapses of the videos are not related with any activity of interest. The very competitive result is reported in the appendix of the second version of paper, which demonstrated the generalization and robustness of our V4D. In fact, unlike previous video compositional methods, even when local events are not well aligned or misclassified, long-term modelling with 4D convolution and video-level aggregation with global average pooling are very likely to correct the partial error.
>
>
> 3.About complexity, in the original version of the paper, we have reported parameters and FLOPs of V4D and compared it with other baseline methods in Table 2.
>
>
> 4. We have already corrected the typo in title in the second version of the paper. Yet it seems that we are not able to modify the title on OpenReview. Thank you for pointing it out.
>
> Hopefully our rebuttal could stress your concerns. If there are still any possible issues, please don’t hesitate to tell us and we will response as soon as possible.

---

### Official Review · AnonReviewer3 · 2019-10-23
**Official Blind Review #3**

**Rating:** 6

**Review:**

The paper proposes 4d convolution and an enhanced inference strategy to improve the feature interaction for video classification.  State-of-the-art performance is achieved on several datasets.  However, there are still details and concerns.

1. The paper should also talk about the details of ARNet and discuss the difference, as I assume they are the most related work
2. during sampling, either training or testing, how do authors handle temporal overlap or make it overlap?
3. can you provide the training memory, inference speed, and total training time?
4. Most importantly, for table4, authors are comparing to the ResNet18 ARNet instead of ResNet50? which is better than the proposed method.
5. lack of related work:  4D Spatio-Temporal ConvNets: Minkowski Convolutional Neural Networks, CVPR 2019.



**Experience Assessment:**

I have published in this field for several years.

**Review Assessment: Checking Correctness Of Derivations And Theory:**

I carefully checked the derivations and theory.

**Review Assessment: Checking Correctness Of Experiments:**

I carefully checked the experiments.

**Review Assessment: Thoroughness In Paper Reading:**

I read the paper at least twice and used my best judgement in assessing the paper.

---

> ### Author Response · Authors · 2019-11-13
> **Response for Reviewer#3**
>
> Thank you for your comments and suggestions. We will address the issues you mentioned.
>
> 1.	ARTNet is not very much related to our V4D. Basically, ARTNet is an alternative for 3D CNNs by replacing 3D convolution layers with SMART blocks. The SMART blocks are two branch units, with one branch for learning static appearance features and one branch for learning motion features. ARTNet is a clip-based method for learning short-term representations while our V4D is a video-level method for learning both short-term and long-term representations.
>
>
> 2.	During training, we uniformly divide the whole video into U sections and randomly select one action unit from each section. So there are no overlaps for training.
> For testing, there might be overlapping during sampling due to the limit length of video. However, our V4D inference algorithm in section 3.4 guarantees that only the non-overlapping action units will interact with each other during testing.
>
>
> 3.	Sure. We train all the models with 8 GPUs of GTX 1080 with memory capacity of 11178MB. For the inference speed, our V4D ResNet18 takes 0.67s per video and V4D ResNet50 takes 1.22s per video. In addition, we also reported the GFLOPs of V4D and compared it with other typical methods in Table 2(b). For training on Mini-Kinetics, V4D ResNet18 takes a bit more than 1 day while V4D ResNet50 takes a bit more than 3 days.
>
>
> 4.	We can only find the results of ARTNet ResNet18 in the published paper. After communication with the authors of ARTNet, we confirm that there are no results published for ARTNet ResNet50. So instead we implement ARTNet ResNet50 by ourselves and the top1 accuracy on Kinetics-400 is 74.3%. This is still lower than our V4D ResNet50 whose top1 on Kinetics-400 is 77.4%. Also, ARTNet ResNet18 reports an average metric of 81.4%, which is the average of top1 and top5 accuracy. While our V4D yields an average score of 85.3%.
>
>
> 5.	Thank you for providing this related work and we now cite this paper in the second version. This paper utilizes 4D CNN to process videos of point cloud so that their input is 4D data. Instead, our V4D processes videos of RGB frames so that our input is 3D data. This basically makes the methods and tasks quite different. Actually we was going to cite this paper yet considering the significant difference we finally did not cite it in the original version.
>
>
> Hopefully our rebuttal could stress your concerns. If there are still any possible issues, please don’t hesitate to tell us and we will response as soon as possible.

---

### Official Review · AnonReviewer1 · 2019-10-23
**Official Blind Review #1**

**Rating:** 3

**Review:**

This paper presents 4D convolutional neural networks for video-level representations. To learn long-range evolution of spatio-temporal representation of videos, the authors proposed V4D convolution layer. Benchmark on several video classification dataset shows improvement.

1. In section 3.1, the authors selected a snippet from each section, but this was not rigorously defined. Same for action units. It does intuitively makes sense, but more mathematical definition (e.g., dimensionality) may be needed.

2. In section 3.2, the authors argued that 3D kernel suffers from trade-off between receptive field and cost of computation. At the end of the subsection, the authors argue that 4D convolution is just k times larger than 3D kernels, which sounds like contradicting. 3D convolution is already expensive and not scalable, but 4D operation sounds even more expensive and more prohibitive.

3. In the paper, the authors argued that clip-level feature learning is limited as it is hard to learn long-range spatio-temporal dependency. It makes sense, and I expect the proposed model may benefit from its design for long-range spatio-temporal feature learning. However, what I see in the experiments is on ~300 frames for Mini-Kinetics and 36-72 frames for Something-Something dataset. Assuming that a second is represented with 15-30 frames, this corresponds to 10-20 sec and 1-4 sec, respectively. I'd say these short videos are still clips.

The paper presents an interesting idea, but there are some issues that need to be addressed before published on ICLR.

**Experience Assessment:**

I have read many papers in this area.

**Review Assessment: Checking Correctness Of Derivations And Theory:**

N/A

**Review Assessment: Checking Correctness Of Experiments:**

I assessed the sensibility of the experiments.

**Review Assessment: Thoroughness In Paper Reading:**

I made a quick assessment of this paper.

---

> ### Author Response · Authors · 2019-11-13
> **Response for Reviewer#1**
>
> Thank you for your comments and suggestions. We will address the issues you mentioned.
>
>
> 1.	We have added more details about sampling strategy to section 3.1 in the new version, with mathematical definition and dimensionality explicitly described.
>
>
> 2.	We did not argue the computation cost of 3D kernels in section 3.2. Instead, we argued that 3D kernels usually are not large enough to cover the holistic video so that Max Pooling operations are applied in most 3D CNNs to enlarge the receptive field. Yet this causes the loss of detailed information. But indeed, in order to preserve the details and increase the receptive field, simply enlarge the 3D kernels to cover the holistic video will bring enormous computation cost. Considering a video of size UxTxHxW, where U is number of action units, and T,H,W means temporal length, height and width of each action unit. In order to model the interaction of 1st frame and the (kT+1)th frame, a 3D kernel of at least (kT+1) x k x k has to be applied, which brings linear increasing of computation cost. Yet with our 4D kernels, a simple k x k x k x k will cover the interaction from the 1st frame to the (kT+1)th frame, because 4D convolution can go beyond space and time, making the long range interaction possible.
>
> For parameters, 4D kernels are k times larger than 3D kernels. So in order to reduce the parameters, we apply k x k x 1 x 1 kernels in most experiments, as mentioned in section 3.2 and section 4.2. We also propose Residual 4D Blocks to ease the optimization and preserve short-term details.
>
>
> 3.	Yes, Mini-Kinetics and Kinetics contain videos about 10 seconds. For Something-Something-v1, they select one frame from every 12 frames so that the original video should be around 430 frames to 860 frames, which are of about half or one minute. We agree that these are still too short to be called videos. Yet here we call our method “video-level” mainly to stress that our V4D models the holistic duration instead of a certain part. Additionally, we evaluated our V4D for untrimmed video classification on ActivityNet v1.3, which contains videos of 5 to 10 minutes and typically large time lapses of the videos are not related with any activity of interest. The very competitive result is now reported in the appendix of the second version of paper.
>
> Hopefully our rebuttal could stress your concerns. If there are still any possible issues, please don’t hesitate to tell us and we will response as soon as possible.

---

### Author Response · Authors · 2019-11-15
**Updated version with untrimmed video classification and visualization**

We have updated a third version of the paper. In the appendix, we further add visualization results and compare V4D with 3D TSN by implementing 3D Class Activation Maps.

---

### Decision · Program_Chairs · 2019-12-19

**Decision:**

Accept (Poster)

**Comment:**

This paper proposes video-level 4D CNNs and the corresponding training and inference methods for improved video representation learning. The proposed model achieves state-of-the-art performance on three action recognition tasks.
Reviewers agree that the idea well motivated and interesting, but were initially concerned with positioning with respect to the related work, novelty, and computational tractability. As these issues were mostly resolved during the discussion phase, I will recommend the acceptance of this paper. We ask the authors to address the points raised during the discussion to the manuscript, with a focus on the tradeoff between the improved performance and computational cost.